# Effluent Molecular Analysis Guides Liver Graft Allocation to Clinical Hypothermic Oxygenated Machine Perfusion

**DOI:** 10.3390/biomedicines9101444

**Published:** 2021-10-11

**Authors:** Caterina Lonati, Andrea Schlegel, Michele Battistin, Riccardo Merighi, Margherita Carbonaro, Paola Dongiovanni, Patrizia Leonardi, Alberto Zanella, Daniele Dondossola

**Affiliations:** 1Center for Preclinical Research, Fondazione IRCCS Ca’ Granda Ospedale Maggiore Policlinico, 20122 Milan, Italy; michele.battistin@policlinico.mi.it (M.B.); riccardo.merighi92@gmail.com (R.M.); dondossola.daniele@gmail.com (D.D.); 2Hepatobiliary Unit, Careggi University Hospital, University of Florence, 50139 Florence, Italy; schlegel.andrea@outlook.de; 3Swiss HPB and Transplant Center, Department of Visceral Surgery and Transplantation, University Hospital Zurich, 8000 Zurich, Switzerland; 4General and Liver Transplant Sugery Unit, Fondazione IRCCS Ca’ Granda Ospedale Maggiore Policlinico, 20122 Milan, Italy; margherita.carbonaro@unimi.it; 5General Medicine and Metabolic Diseases, Fondazione IRCCS Ca’ Granda Ospedale Maggiore Policlinico, Via Francesco Sforza 35, 20122 Milan, Italy; paola.dongiovanni@policlinico.mi.it; 6Department of Pathophysiology and Transplantation, University of Milan, 20122 Milan, Italy; patrizia.leonardi@unimi.it (P.L.); alberto.zanella1@unimi.it (A.Z.); 7Department of Anesthesia and Critical Care, Fondazione IRCCS Ca’ Granda Ospedale Maggiore Policlinico, 20122 Milan, Italy

**Keywords:** hypothermic-oxygenated-machine-perfusion (HOPE), liver transplantation (LT), brain death donors (DBD), extended criteria donors (ECD), donation after circulatory death donors (DCD), effluent fluids, molecular profile, data-driven decision making, early allograft dysfunction (EAD), hyaluronan (HA)

## Abstract

Hypothermic-oxygenated-machine-perfusion (HOPE) allows assessment/reconditioning of livers procured from high-risk donors before transplantation. Graft referral to HOPE mostly depends on surgeons’ subjective judgment, as objective criteria are still insufficient. We investigated whether analysis of effluent fluids collected upon organ flush during static-cold-storage can improve selection criteria for HOPE utilization. Effluents were analyzed to determine cytolysis enzymes, metabolites, inflammation-related mediators, and damage-associated-molecular-patterns. Molecular profiles were assessed by unsupervised cluster analysis. Differences between “machine perfusion (MP)-yes” vs. “MP-no”; “brain-death (DBD) vs. donation-after-circulatory-death (DCD)”; “early-allograft-dysfunction (EAD)-yes” vs. “EAD-no” groups, as well as correlation between effluent variables and transplantation outcome, were investigated. Livers assigned to HOPE (*n* = 18) showed a different molecular profile relative to grafts transplanted without this procedure (*n* = 21, *p* = 0.021). Increases in the inflammatory mediators PTX3 (*p* = 0.048), CXCL8/IL-8 (*p* = 0.017), TNF-α (*p* = 0.038), and ANGPTL4 (*p* = 0.010) were observed, whereas the anti-inflammatory cytokine IL-10 was reduced (*p* = 0.007). Peculiar inflammation, cell death, and coagulation signatures were observed in fluids collected from DCD livers compared to those from DBD grafts. AST (*p* = 0.034), ALT (*p* = 0.047), and LDH (*p* = 0.047) were higher in the “EAD-yes” compared to the “EAD-no” group. Cytolysis markers and hyaluronan correlated with recipient creatinine, AST, and ICU stay. The study demonstrates that effluent molecular analysis can provide directions about the use of HOPE.

## 1. Introduction

The adoption of expanded criteria for acceptance of organ donors broadened the pool of livers available for transplantation. However, relative to standard livers, organs procured from either brain death (DBD) extended criteria (ECD) or donation after circulatory death (DCD) donors are associated with enhanced risk of postoperative complications and transplant failure [1,2,3]. In this perspective, the machine perfusion (MP) technique addresses the clinical need to evaluate viability and function of marginal grafts before transplantation [1,4,5,6,7,8,9,10,11,12]. 

In clinical practice, the decision to accept a marginal graft and to perform ex situ perfusion is based on both donor and graft characteristics, including occurrence of steatosis, cold ischemia time (CIT), warm ischemia time (WIT), donor brain or cardiac death, donor age, and donor comorbidities [13]. Although different indexes are available to assess donor factors influencing graft survival [14,15,16,17,18], their utility in guiding clinicians about MP referral is debated. In fact, generalization of such scores is hampered by regional differences in criteria for donor acceptance (e.g., donor age in Italy), in laws governing organ donation, in assessment of donor–recipient matching, and in allocation systems [14]. Thus, due to insufficient objective criteria able to assess the need to perform ex situ perfusion, clinicians’ judgment remains crucial in the decision making on MP use [13]. 

The analysis of effluent fluid, which is released upon organ flush on the bench and includes the residual preservation solution, could provide useful information to implement selection criteria for the livers that might benefit from further evaluation during MP. In fact, both clinical [19,20,21,22,23,24,25,26,27,28,29,30,31] and preclinical [32,33,34,35,36] studies showed that effluent concentration of different parameters correlates with preservation injury and early graft performance. More specifically, cytolysis enzymes [19,20,23,24,25,29,31,33,36]—i.e., aspartate aminotransferase (AST), alanine transaminase (ALT), lactate dehydrogenase (LDH), alkaline phosphatase (ALP)-, lactate [23,32,35], and damage-associated molecular patterns (DAMPs) [22,28,29,37] were identified as biomarkers of liver injury and predictors of transplantation outcome. Moreover, the evaluation of inflammatory mediator concentration in liver effluents could help to assess the influence of brain death, donor (d) WIT, and cold ischemia on the graft quality and, consequently, on post-transplant outcome [23,27]. 

We investigated if the assessment of specific factors in effluent fluids can help clinicians with decision making regarding the allocation of marginal graft to MP. In particular, we elected to assess the effluent concentration of biomarkers of hepatocellular damage, inflammation-related mediators, and DAMPs, which have been previously associated with brain death-induced inflammation, liver IRI, and poor transplantation outcome [38,39,40]. The analysis was retrospectively performed in effluent samples collected from both DBD and DCD human livers, either subjected to MP or static cold storage (SCS) and then transplanted. The main focus was to determine if livers referred to ex situ perfusion based on clinicians’ decision were associated with a distinctive molecular profile, compared to grafts that did not undergo MP. In addition, the presence of a peculiar signature in the effluent of DCD grafts was investigated. Finally, we evaluated the potential predictive value of such effluent molecular analysis on the short-term outcome after transplantation.

## 2. Materials and Methods 

### 2.1. Study Design and Effluent Fluid Collection

The present research includes all consecutive adult patients (*n* = 49) who received a liver graft procured from either DBD or DCD donors from February 2017 to May 2019 in our center. Late re-LT and patients with incomplete data were excluded (*n* = 11).

At the end of the back-table procedure (eventually before MP), the cava vein was clamped and grafts were flushed through the portal vein with 1000 mL of Celsior Solution (IGL, Lissieu, France). Effluent samples were collected through a 16G ago-cannula inserted in the posterior portion of the retrohepatic cava vein (Appendix A).

### 2.2. Donors, Liver Procurement, and Preservation

Grafts were categorized as ECD based on the criteria described by Vodkin and coworkers [41]. Donor age was considered as non-standard only if >80 years [42].

The grafts were allocated to our center according to the North Italian Transplant program allocation policy. Our surgical team procured all grafts, with the exception of 4 national or interregional allocated livers [43]. Alternative procedures for liver graft procurement were applied in DCD and DBD donors [44]. Normothermic regional perfusion (NRP) was applied after death declaration and before organ procurement in all DCD donors according to the Italian law. It consists of abdominal extracorporeal membrane oxygenation directly applied in the donors though the placement of femoral cannulas after death declaration [45].

All grafts underwent in situ flushing with Celsior solution (IGL) and static preservation in an ice-box.

### 2.3. Liver Machine Perfusion (MP)

Liver MP was applied to all DCD grafts and to DBD extended-criteria donor (ECD) organs showing one of the following characteristics: expected prolonged ischemia time (>10 h), macrovescicular steatosis >30%, or serum levels of hepatonecrosis markers exceeding 4 times the reference range.

MP was performed as back-to-base, end-ischemic dual hypothermic oxygenated machine perfusion (D-HOPE). Briefly, D-HOPE was performed using the Liver Assist device (Organ Assist, Groningen, The Netherland), and consisted in a double perfusion (4000 mL of circulating Belzer MPS-UW Machine Perfusion) through the portal vein and hepatic artery with pressures set at 4 and 25 mmHg, respectively. Grafts were actively oxygenated (FiO_2_ = 100%) at a temperature of 8–10 °C.

### 2.4. Liver Transplantation (LT) and Classification of Post LT Complications

Liver transplantation (LT) was performed with a modified piggyback technique and bile duct anastomoses over a T-tube. Post-LT course and biochemical analysis were performed (liver and kidney function test, coagulation, blood count).

Early allograft dysfunction (EAD) was defined as: peak transaminases > 2000 IU/L (within the first week after LT), international normalized ratio > 1.7, or bilirubin > 10 mg/dL on postoperative day (POD) 7 [46]. Primary non-function (PNF) was defined as a non-recoverable graft function needing urgent re-LT < 10 POD (AST > 5000 UI/L, INR > 3.0, and metabolic acidosis) with no sign of vascular pathology of the implanted liver [47]. Acute kidney injury (AKI) was defined and staged according to KDIGO criteria; we considered patients as affected by post-LT AKI only if assigned to ≥stage 2 [48].

### 2.5. Effluent Processing and Molecular Analysis

Samples of effluent were processed to obtain cell-free supernatants and donor-derived leukocyte pellets (the protocol is described in detail in [49,50,51]). Effluents collected during back-table procedure before MP were analyzed to measure the concentration of cytolysis enzymes, flavin mononucleotide (FMN), caspase-cleaved keratin 18 (CK18), inflammatory mediators, metabolites, and electrolytes.

#### 2.5.1. Biochemical Tests

A biochemical panel including transaminases (AST, ALT), Blood Urea Nitrogen (BUN), glucose, lactate, LDH, D-dimer, and electrolytes was assessed following standard techniques.

#### 2.5.2. Flavin Mononucleotide (FMN)

The FMN concentration was determined by fluorescence spectroscopy. Briefly, 5 mg of Riboflavin 5′-monophosphate sodium salt hydrate (Sigma-Aldrich, Merck KGaA, Darmstadt, Germany) were dissolved in either 0.9% sodium chloride or Celsior. Thereafter, these solutions were serially diluted to obtain different standard FMN solutions with concentration ranging from 31.25 to 1000.00 ng/mL (Appendix A). Effluent samples were then dispensed in triplicate in black microplates, together with the standard samples. Fluorescence readings were performed using a multi-mode microplate reader (Synergy HTX, Biotek U.S, Winooski, VT, USA). A monochrome light with excitation wavelength of 460/40 nm was used, while fluorescence emission was revealed with 100% gain at 528/20 nm. Finally, the average fluorescence readings of effluent samples were plotted against the appropriate standard curve to calculate sample FMN concentration.

#### 2.5.3. Soluble Caspase-Cleaved Keratin (CK) 18

Hepatocyte apoptosis was assessed by measuring the release of caspase-cleaved CK18 with a commercially available immunoassay based on the use of M30 monoclonal antibody that specifically detects the neoepitope ccK18/K18-Asp396 (M30-Apoptosense ELISA kit, PEVIVA, VLVbio-BioScientific Pty. Ltd., Stockholm, Sweden). The analysis was performed following manufacturers’ instructions and the absorbance was determined using a microplate reader (Synergy HTX).

#### 2.5.4. Mediator Concentration

A panel of soluble proteins relevant to inflammation and its resolution (CCL2/MCP-1, CXCL8/IL-8, CXCL9/MIG, IL-1β, IL-6, PTX3, TNF-α, IL-10, ANGPTL3, ANGPTL4, Galectin-9, HGF), glycocalyx (Glypican, HA, Syndecan-1), coagulation (Protein C/Factor IVX, PAI-1) was evaluated using custom-designed immunoassays based on Luminex xMAP^®^ Technology (R&D Systems, Minneapolis, MN, USA; fluorescence detection by Luminex 200, Austin, CA, USA) or commercial available enzyme-linked immunosorbent assay (ELISA) assays (R&D Systems; absorbance measurement carried out with Synergy HTX). A list of all the measured analytes with low and high standards as well as test sensitivity is provided in the Appendix A.

#### 2.5.5. Leukocyte Count and Free Hemoglobin

Cell pellets were suspended in 0.25 mL 1X PBS solution (Sigma-Aldrich) and counted using an automated cell counter (Scepter, Millipore Corporation, Billerica, MA, USA).

Free hemoglobin concentration was assessed by applying the Allen correction to absorbance readings at 563 nm, 577 nm, 600 nm (Synergy HTX).

### 2.6. Statistical Analysis

Statistical analysis was performed using the JMP Pro 15 software (© SAS Institute Inc., Cary, NC, USA). Box plots and linear regression plots were created using SigmaPlot 11.0 (Systat Software Inc., San Jose, CA, USA).

Continuous variables are presented as median (25–75 or absolute values with percentage, while categorical variables were expressed in percentages. To reduce the distribution effect of the flush solution volume and to normalize data according to liver size, all the measured variables are shown as total release per gram of liver graft.

Multivariate analysis was used to explore associations across all the variables evaluated in effluent samples.

To identify any potential specific signature of effluent parameters, an unsupervised agglomerative hierarchical cluster analysis was performed (NA-chip analyzer program, https://sites.google.com/site/dchipsoft/, accessed on 1 July 2021. Thereafter, Wilcoxon-Rank tests were independently run to investigate if there were significant differences in the following comparisons: “MP-yes” vs. “MP-no”; DCD vs. DBD; “EAD-yes” vs. “EAD-no”; “steatosis-yes” vs. “steatosis-no”; “age < 65” vs. “age > 65”; “age < 80” vs. “age > 80”.

Finally, correlation between effluent variables and transplantation outcome was investigated using the Pearson’s correlation coefficient.

A probability value < 0.05 was considered significant.

## 3. Results

### 3.1. Donor Characteristics and Liver Graft Allocation to the MP Procedure

Donor data and procurement timings are shown in Table 1. Ten (20%) grafts were derived from DCD donors (7 category III and 3 category II) and 39 (80%) from DBD donors. The total preservation time lasted 565 min (510–660). All DCD liver grafts showed a high-risk profile, according to the UK-DCD score [18]. Consistently, their median DRI was 2.21 (1.85–2.37), which was higher than the DRI of 1.6 observed in DBD livers. In addition, six grafts showed a macrovescicular steatosis of >30%.

Eighteen (47%) grafts were subjected to MP, where 10 were procured from DCD and 8 from DBD-ECD donors meeting the criteria specified above, except for livers 18 and 19, which were subjected to MP to extend the preservation time because of logistic reasons (change of recipient) [36,45,52].

### 3.2. Recipients’ Characteristics and Outcome of Transplantation

The mean recipient model for end stage liver disease (MELD) score was 13 (9–17) points, while 11% of the patients received an urgent allocated graft (Table 1). Pre-LT chronic renal failure was diagnosed in four (11%) cases. One-year graft survival was 90% (44/49) with a patient survival of 92% (45/49). Two recipients died (sepsis, intracranial hemorrhage) within 10 days after transplantation. One patient was re-transplanted because of a hepatic artery thrombosis.

Despite an overall high-risk population in our cohort, graft survival, censored for tumor-related death, remained comparable to the levels seen in an ideal LT cohort presented in a recent benchmark study (Table 1, no cases of PNF, 1 case of ischemic type biliary lesion in a DBD graft) [53]. EAD was diagnosed in 7 cases and 14 patients developed AKI.

### 3.3. Molecular Analysis of Effluent Fluids

#### 3.3.1. Associations across the Measured Parameters

Different correlations across the variables assessed in effluent fluids were found through multivariate analysis (Appendix A). The following molecule patterns were identified (*p* < 0.05): (1) TNF-α, IL-1β, CXCL8/IL-8, CXCL9/MIG; (2) ANGPTL3, ANGPTL4, Galectin-9, HGF; (3) AST, ALT, LDH.

Of note, IL-10 was inversely associated with TNF-α (r = −0.653, *p* = 0.040), ANGPTL3 (r = −0.551, *p* = 0.001), ANGPTL4 (r = −0.547, *p* = 0.003), and CXCL9 (r = −0.546, *p* = 0.001). HA was associated with both inflammatory mediators (TNF-a: r = 0.869, *p* = 0.001; IL-1β: r = 0.427, *p* = 0.042; CXCL9/MIG: r = 0.556, *p* = 0.006) and markers of hepatocyte injury (AST: r = 0.689, *p <* 0.001; ALT: r = 0.583, *p* = 0.003; LDH: r = 0.533, *p* = 0.001; FMN: r = 0.517, *p* = 0.001). The apoptosis marker CK18 correlated with CCL2 (r = 0.510, *p* = 0.030), HGF (r = 0.711, *p* = 0.001), IL-6 (r = 0.748, *p <* 0.001), and free hemoglobin (r = 0.640, *p* = 0.034). Finally, a strong association was observed between FMN and free hemoglobin (r = 0.781, *p <* 0.001).

#### 3.3.2. Effluents of Livers Referred to MP Procedure Show a Peculiar Molecular Signature

Unsupervised clustering analysis disclosed a distinctive signature in the bio-molecular profile of livers referred to MP procedure. In fact, half of the grafts subjected to MP (livers 35, 37, 30, 2, 14, 8, 32, 39, 41; 9/18, 50%) were grouped together due to their similar profiles (Figure 1, blue dendrogram, *p* = 0.021). Out of the nine grafts excluded from the “MP-yes” cluster, six were procured from DCD donors (livers 38, 34, 40, 16, 7, 42; 6/9), while three were from DBD donors (livers 29, 18, 36; 3/9). Of note, three livers were grouped in a distinctive sub-cluster due to high expression of inflammation-related molecules (Figure 1, livers 38, 34, and 40, *p* = 0.018).

Consistently, statistical analysis indicated substantial differences between effluents collected from livers referred to MP compared to those of the grafts immediately used for transplantation. In particular, the “MP-yes” group was associated with increased effluent concentrations of the inflammatory molecules PTX3, CXCL8/IL-8, TNF-α, and ANGPTL4 (Figure 2). Conversely, fluids recovered from livers assigned to ex situ perfusion showed lower concentrations of FMN and IL-10 (Figure 2). Finally, there was no difference in the effluent concentration of the cytolysis enzymes AST (“MP-no” vs. “MP-yes”: 0.26 (0.21–0.44) IU/g vs. 0.35 (0.27–0.66) IU/g, *p* = 0.419), ALT (0.22 (0.20–0.38) IU/g vs. 0.42 (0.18–0.70) IU/g, *p* = 0.260), and LDH (0.52 (0.35–0.86) IU/g vs. 0.64 (0.40–1.34) IU/g, *p* = 0.437).

#### 3.3.3. Differences in the Bio-Molecular Profiles of Effluent Fluids Collected from DCD and DBD Livers

Data reported in Table 2 indicate that livers procured from DCD and DBD donors show different effluent molecular profiles. Among differentially expressed mediators, there were molecules relevant to inflammation (CCL2/MCP-1, CXCL8/IL-8, CXCL9/MIG, TNF-α, ANGPTL4), hepatocyte apoptosis (CK18), and coagulation (Protein C, D-Dimer). Of note, concentration of the anti-inflammatory cytokine IL-10 was lower in effluent fluids of DCD compared to those obtained from DBD livers. Differences in BUN and K+ levels were likewise observed.

Finally, no relationship was found between the concentration of lactate, AST, and ALT assessed in NRP perfusates and the release of these factors in effluent fluids.

### 3.4. Association with Transplantation-Related Complications and Recipient Outcome

Effluent concentration of cytolysis markers AST, ALT, and LDH was significantly higher in the “EAD-yes” group (Figure 3). Conversely, there was an increased release of K+ in the samples from the “EAD-no” group (0.012 (0.010–0.012) mmol/g vs. 0.015 (0.012–0.016) mmol/g, *p* = 0.016). With regard to the other biomarkers assessed in effluent fluids, no significant difference was observed between the “EAD-yes” and the “EAD-no” groups (Appendix A).

A further analysis investigated any potential association between the variables measured in the effluent fluids and clinical data related to transplantation outcome, including days of recipient hospitalization in intensive care unit (ICU) and biochemical parameters of liver and kidney function assessed in recipients’ plasma samples (peak). Effluent biomarkers of liver damage correlated with recipient serum creatinine (LDH: r = 0.517, *p* = 0.001; AST: r = 0.538, *p* = 0.0007; ALT: r = 0.570, *p* = 0.0003), recipient serum AST (LDH: r = 0.745, *p <* 0.0001; AST: r = 0.726, *p <* 0.0001; ALT: r = 0.698, *p <* 0.0001), and ICU stay (LDH: r = 0.459, *p* = 0.001; AST: r = 0.400, *p <* 0.0001; ALT: r = 0.381, *p* = 0.029). Of interest, strong or moderate correlations were found between effluent HA and recipient creatinine, AST, and ICU stay (Figure 4). Finally, there were weaker positive associations between ICU stay and effluent concentration of ANGPTL4 (r = 0.532, *p* = 0.009) and CXCL8 (r = 0.329, *p* = 0.048).

## 4. Discussion

The present proof-of-concept study indicates that the analysis of effluent fluids collected from liver grafts during SCS can provide valuable information to guide surgeons in the decision making on D-HOPE application to marginal organs. In fact, the clinical decision to allocate specific livers to the MP procedure was retrospectively supported by a peculiar molecular profile in the liver effluents. In addition, a donor-specific signature was observed, with increased concentration of inflammation- and cell death-related factors in the effluent fluids collected from DCD grafts compared to those seen in DBD livers. Finally, an interesting correlation between the effluent concentrations of certain molecules and EAD as well as recipient clinical data was observed.

Higher numbers and quality of donor organs can be lifesaving for many patients waiting for a transplant. The MP technique allows a possible evaluation of viability and function as well as reconditioning of suboptimal grafts before transplantation. However, there are no reliable criteria to determine whether a marginal liver is immediately suitable for transplantation or if it rather might benefit from further treatment with MP. Therefore, the decision to perfuse a specific liver or not is currently based on a few objective criteria, on the center-specific policy, and on surgeons’ individual “gut feeling” [13].

The assessment of specific parameters in effluent fluids could reduce the uncertainty about liver viability and would facilitate the preoperative estimation of graft quality. In fact, such biological material has a huge diagnostic potential as it contains molecules and cells from the graft, challenged by the noxious stimuli inherent in the transplantation procedure, from the donor treatment withdrawal and surgery throughout the entire preservation period until implantation. Of note, the effluent analysis provides information on the entire liver parenchyma rather than on a small part of the hepatic tissue, which is a clear advantage compared to liver biopsies. Consistently, several researchers demonstrated that the effluent composition can reveal both pre-existing injury, including brain death- and steatosis-related inflammation, and the damage acquired during procurement and preservation [19,20,21,22,23,24,25,26,27,28,29,30,31,32,33,34,35].

We therefore hypothesized, that the molecular analysis of effluents could help with the selection of marginal livers, that may require a more accurate assessment and reconditioning with MP. To test this idea, we performed an analysis of effluent samples collected from grafts either subjected to MP based on the surgeon’s judgment or transplanted without this procedure. In addition to cytolysis enzymes and metabolites, we investigated the concentration of inflammation-related mediators and DAMPs, previously shown to be accurate markers of brain death-induced inflammation and damage, liver IRI, and poor transplantation outcome [38,39,40].

Our study discloses a distinctive molecular signature in the effluents of livers referred to the D-HOPE procedure. The most remarkable observation is the reduced concentration of the anti-inflammatory mediator IL-10 in fluids of grafts, which underwent MP, compared to those collected from livers immediately used for transplantation. IL-10 is a pleiotropic anti-inflammatory cytokine expressed in different liver cell types, including hepatocytes, Kupffer cells, sinusoidal endothelial cells, type 2 T-helper (Th2) cells, and lymphocytes [54]. Of note, recent evidence indicated IL-10 as a master regulator of macrophage efferocytosis and phenotypic conversion in sterile liver injury, such as IRI [55]. In particular, this molecule induces macrophage switching from a proinflammatory to a restorative phenotype and a low concentration of IL-10 was associated with failure of debris uptake, prolonged liver damage, aberrant hepatocyte proliferation and fibrosis [55,56]. Another interesting result of our research concerns the higher release of inflammation-related mediators, such as PTX3 and CXCL8/IL-8, from livers subsequently subjected to MP. Beside activated leukocytes, DAMPs and inflammatory mediators are released by stressed and dying hepatocytes. IL-8/CXCL-8 promotes neutrophil and macrophage migration to the liver and regulates hepatocyte survival and proliferation upon IR injury [57]. Of note, increased serum IL-8/CXCL-8 concentration was associated with higher serum transaminases in patients who received a liver transplantation [27]. PTX3 is a pattern-recognizing protein produced by macrophages and hepatic stellate cells in response to proinflammatory signals and Toll-like receptor engagement [58]. In liver pathology, the detection of an elevated blood concentration of PTX3 is an index of hepatic disease [38,59]. These observations collectively show that in our case series, livers subjected to MP were actually associated with a pro-inflammatory and non-resolutive effluent profile compared to grafts not referred to D-HOPE. Interestingly, clinical [8] and preclinical [60] studies showed that the application of D-HOPE can efficiently modulate such inflammatory and cell death pathways activated by perimortem and donation events and exacerbated by high-risk donor characteristics. The beneficial effect of D-HOPE is based on the induction of mitochondrial Complex I and II reprogramming [10,61], with subsequent improved function of the entire respiratory chain, improved Succinate metabolism, and enhanced ATP production. This ultimately leads to reduced reperfusion injury and lower risk of post-LT complications [10].

With regard to traditional markers of liver injury, there was no significant difference in effluent concentration of AST, ALT, and LDH of livers referred to MP, compared to grafts not subjected to ex situ perfusion. Of note, the present study includes the evaluation of FMN, a novel biomarker of cell viability, whose concentration is measured in perfusate samples to assess liver graft mitochondrial function throughout HOPE [62]. Although increasing data support its robustness and reliability in the prediction of organ viability during oxygenated ex situ perfusion [62,63], our results suggest that the evaluation of effluent FMN during SCS could provide different information, probably due to the absence of an active oxygenation and dynamic perfusion. Of interest, the multivariate analysis indicated a positive association between FMN and free hemoglobin concentration. Targeted studies are needed to determine the biological significance of FMN release during donor management and the preservation phase.

Additional analysis documented a peculiar inflammation, cell death, and coagulation profile in fluids collected from DCD livers, compared to those obtained from DBD grafts. This condition could reflect the detrimental influence of prolonged WIT secondary to the 20 min no-touch period, which is mandatory in the Italian setting [45]. Moreover, the marked inflammatory signature observed in our DCD grafts could be a consequence of the NRP procedure, consisting of an oxygenated in situ perfusion, applied following cardiac arrest in DCD organ donation. In fact, in other case-series, where livers were procured immediately after death declaration without NRP [64,65], DCD grafts showed increased necrosis-related biomarkers compared to DBD organs, but no sign of enhanced inflammation. Our data further support the diagnostic power of effluent analysis in liver transplantation, suggesting that it could be used to assess both the degree and the nature of injury in each graft type.

Finally, correlations between recipient short-term outcome and effluent concentration of either cytolysis biomarkers or HA were observed. Along with biliary complications and AKI, EAD is a post-LT complication [66] mainly consequent to IRI-induced hepatocellular injury, endothelial damage, and inflammatory cell activation [67]. In our case series, all evaluated markers of hepatocellular damage correlated with EAD, consistent with previous reports [19,20,23,24,25,29,31,33]. In addition, effluent enzyme levels were positively associated with recipient parameters of liver and kidney function and hospitalization duration. Concerning HA, significant correlations were found between its effluent concentration and recipient blood creatinine, AST and ICU stay. HA is normally taken up from the circulation and metabolized by hepatic microvascular endothelial cells. Therefore, increased effluent HA concentration can denote a loss of integrity or functional failure of sinusoidal cells. Our results are consistent with previous studies, where HA uptake measured in the caval effluent during recipient operation [22,37] or in back-table effluents [29] was lower in patients with poorer early graft function following LT.

A significant limitation in the present study resides in the small number of liver transplantations performed in our center, which limits the validation of each of the variables assessed in effluent fluids. In addition, based on the excellent outcomes with a low number of complications, despite the high donor risk, we were not able to correlate effluent molecular analysis with recipient outcomes. Targeted studies are required to better characterize the prognostic ability of effluent biomarkers to predict the occurrence of post-transplant complications.

## 5. Conclusions

The present proof-of-concept study shows that molecular analysis of effluent samples can provide objective criteria reflecting the cumulative effects of noxious stimuli affecting liver graft quality during the organ donation process. We suggest that bio-molecular stratification of livers would significantly contribute to implement the clinical strategy with or without application of the MP procedure, leading to improved, cost-saving, and tailored healthcare resource utilization based on the specific injury experienced by each graft. In addition, marginal organs that are currently deemed unsuitable because of the presumed high-risk to develop complications could be successfully used in case a “healthy” biomarker profile is found. Of interest, novel technologies can enable surgeons to perform real-time evaluation of effluent fluids during back table preparation [68,69], with results available in time to support the decision making of the surgeon and the entire surgical team. This perspective could pave the way for the adoption of a personalized medicine-based approach in the field of organ transplantation, which would ultimately lead to improved donor-recipient matching and to better post-transplant results.

## Figures and Tables

**Figure 1 biomedicines-09-01444-f001:**
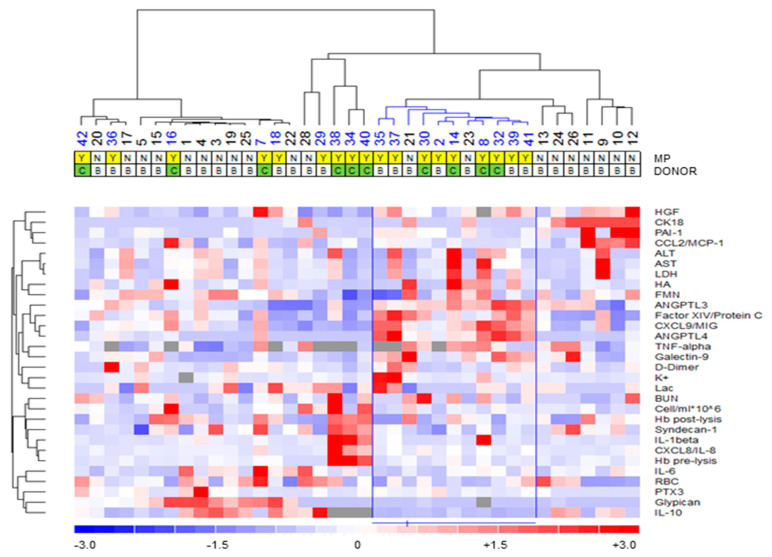
**Agglomerative hierarchical clustering of molecular effluent profile of liver grafts.** Unsupervised analysis was performed in order to investigate whether specific signature could be identified in the effluent fluids. A distinctive profile was recognizable, with 9 out of 18 grafts subjected to MP grouped in a single cluster (*p* = 0.021). Cluster analysis was performed using dCHIP software (clustering method: average linkage; distance metric: 1—Spearman’s rank correlation). Columns identify liver grafts, while rows denote the parameters evaluated in effluent fluids. Y denotes “MP-yes”, while N denotes “MP-no”. Cause of death was referred to as D for DBD donors and C for DCD. The degree of color saturation reflects magnitude of variable concentration, as indicated in the color scale. ALT, alanine transaminase; ANGPTL3, Angiopoietin-like Protein-3; ANGPTL4, Angiopoietin-like Protein-4; AST, aspartate aminotransferase; CCL2/MCP-1, C-C motif chemokine ligand 2/Monocyte chemoattractant protein 1; CK18, caspase-cleaved keratin 18; CXCL8/IL-8, C-X-C Motif Chemokine Ligand 8/Interleukin 8; CXCL9/MIG, C-X-C Motif Chemokine Ligand 9; FMN, flavin mononucleotide; HA, hyaluronan; HGF, Hepatocyte Growth Factor; IL-10, interleukin 10; IL-1β, interleukin 1β; IL-6, interleukin 6; LDH, lactate dehydrogenase; PTX3, Pentraxin 3; TNF-α, Tumor Necrosis Factor α.

**Figure 2 biomedicines-09-01444-f002:**
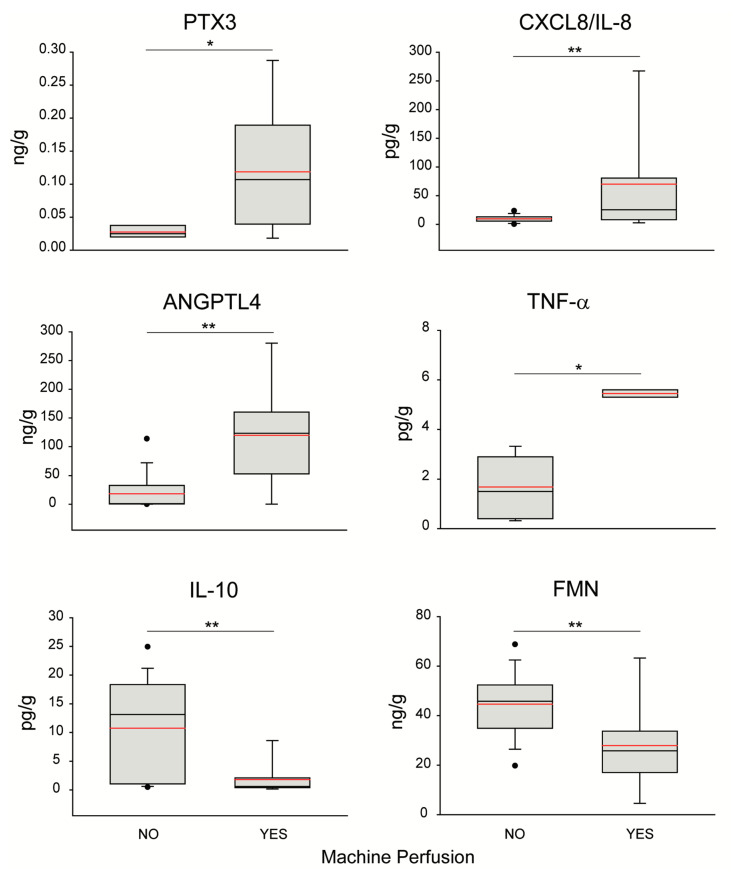
**Differences in relevant variable concentration measured in effluent fluids collected from livers referred to the MP procedure compared to those retrieved from grafts transplanted without MP.** Boxes denote median 5th–95th percentiles; red lines highlight the mean of each study group, while black points denote outliers. Statistical significance was investigated using Wilcoxon non-parametric test. *p* value: * <0.05; ** <0.01. PTX3, pentraxin-related protein 3; CXCL8/IL8, chemokine (C-X-C motif) ligand 8 or interleukin 8; BUN, blood urea nitrogen; FMN, flavin mononucleotide.

**Figure 3 biomedicines-09-01444-f003:**
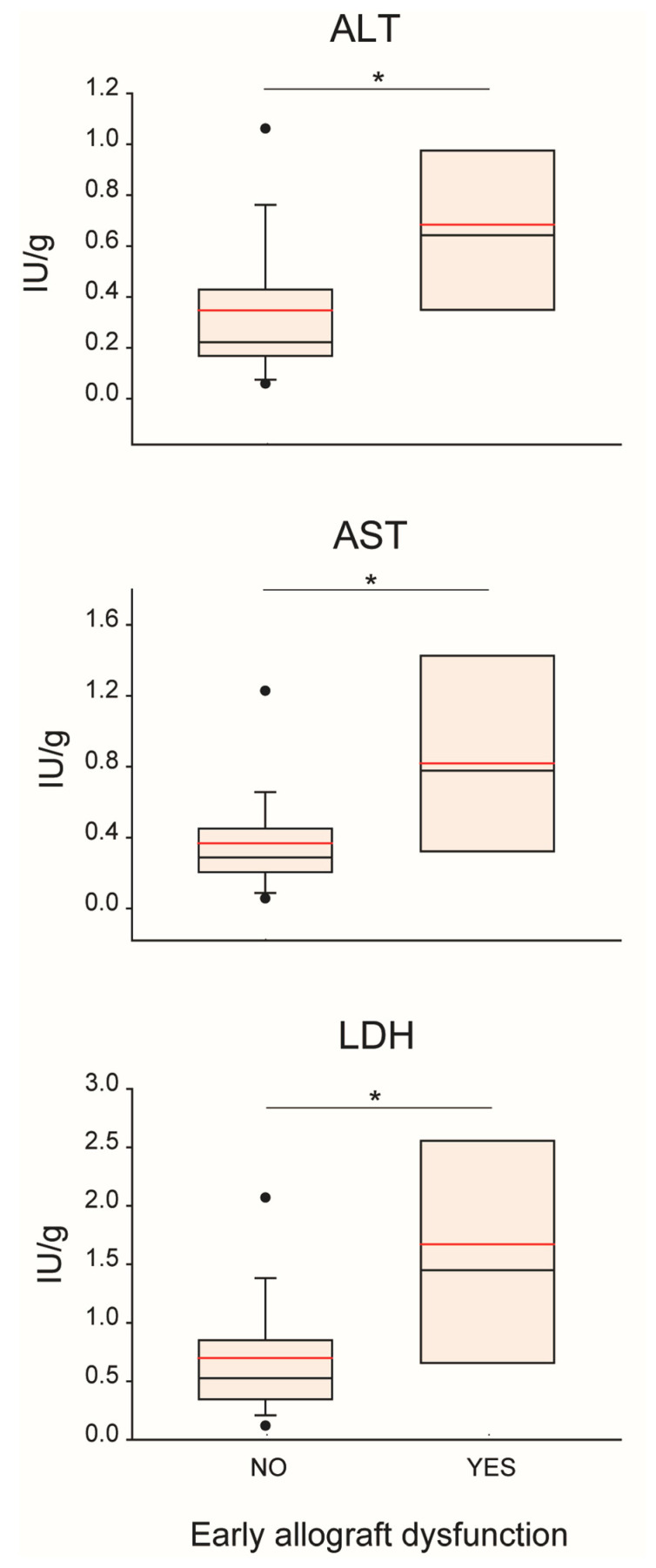
**Effluent cytolysis enzymes according to Early Graft Function (EAD) occurrence.** Patients that developed EAD after LT showed increased effluent concentration of biomarkers of hepatocellular lysis. Boxes denote median 5th–95th percentiles; red lines highlight the mean of each study group, while black points denote outliers. Statistical significance was investigated using Wilcoxon non-parametric test. *p* value: * <0.05. ALT, alanino transferase; AST, aspartate amino transferase; LDH, lactate dehydrogenase.

**Figure 4 biomedicines-09-01444-f004:**
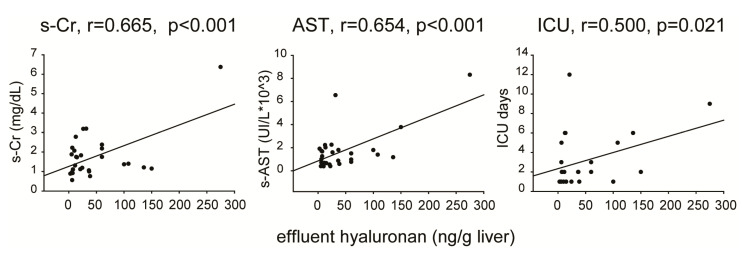
**Linear regression analysis between effluent HA concentration and recipient parameters.** s-CR, serum creatinine; s-AST, serum aspartate amino transferase; ICU, intensive care unit stay.

**Table 1 biomedicines-09-01444-t001:** **Donor and recipient characteristics.** Data are presented as median (25–75) or *n* (%), *n* = 49. Abbreviations: BMI, body mass index; ICU, Intensive care unit stay; DCD, cardiac death donors; DBD, brain death donors; WIT, warm ischemia time; cDCD, controlled or type III DCD; uDCD, uncontrolled or type II DCD; DHOPE, dual hypothermic oxygenated machine perfusion; NRP, normothermic machine perfusion; ECD, extended criteria donors; MELD, model for end-stage liver disease; HCC, hepatocellular carcinoma; PNF, primary non function; EAD, early allograft dysfunction; AKI, acute kidney injury; AST, aspartate aminotransferase; ALT, alanine transferase; re-LT, liver re-transplantation; * only for DCD donor.

Population	Parameter	Median (25–75) or N (%)		
		Overall	DBD Graft	DCD Graft
**Donors**		***n* = 49**	***n* = 39**	***n* = 10**
	Age, years	62 (54–72)	64 (56–75)	55 (53–58)
	Male gender, *n* (%)	28 (57)	18 (46)	10 (100)
	BMI, Kg/cmq	26 (24–27)	26 (24–27)	25 (24–26)
	ICU, days	3 (1–7)	4 (1–7)	2 (1–6)
	Cardiac arrest, *n* (%)	18 (37)	8 (20)	10 (100)
	Cause of death:			
	-DCD, *n* (%)	10 (20)	-	-
	-DBD, *n* (%)	39 (80)	-	-
	Type III DCD, *n*			7 (70%)
	Functional WIT (cDCD), min			40 (32–54)
	Low flow time (uDCD), min			85–87–110
	Total WIT (uDCD + cDCD), min			56 (37–65)
	Macrosteatosis > 20%. *n* (%)	18 (37)	11 (28)	7 (70)
	Macrosteatosis > 30%. *n* (%)	6 (12)	3 (8)	3 (30)
	Machine perfusion, *n* (%)	18 (47%)	8 (20)	10 (100)
	Timing, min:			
	-clamp-effluent collection	330 (200–420)	328 (256–397)	285 (224–367)
	-cold preservation	565 (510–660)	568 (511–653)	565 (530–605)
	Machine perfusion, *n* (%)	18 (47)	8 (20)	10 (100)
	DHOPE duration, min	240 (190–290)	250 (180–290)	240 (180–270)
	NRP duration, min *			335 (280–395)
	Graft weight, g	1545 (1250–1700)	1420 (1255–1700)	1605 (1415–1655)
	ECD, *n* (%)	32 (65)	22 (56)	10 (100)
	Donor-Risk-Index:	1.67 (1.48–2.12)	1.63 (1.43–1.81)	2.21 (1.85–2.37)
	UK DCD score			11 (10–14)
**Recipients**				
	Male gender, *n* (%)	39 (80)	30 (77)	9 (90)
	Age, years	57 (54–63)	57 (53–61)	61 (60–64)
	MELD	13 (9–17)	14 (9–15)	10 (9–15)
	BMI, Kg/cmq	25 (22–27)	25 (23–28)	25 (23–27)
	HCC, *n* (%)	26 (53)	16 (15)	8 (80)
	ICU, days	2 (1–5)	2 (1–3)	3 (1–8)
	Hospital stay, days	17 (14–27)	17 (14–24)	19 (13–25)
	PNF, *n* (%)	0	0	0
	EAD, *n* (%)	10 (20)	7 (18)	3 (30)
	AKI peak, *n* (%)	19 (39)	15 (38)	4 (40)
	AST peak, U/L	1004 (652–1787)	883 (638–1787)	1389 (935–1857)
	ALT peak, U/L	729 (440–1320)	749 (399–1229)	717 (551–1766)
	Re-LT, *n* (%)	1 (2)	1 (2)	0
	Death, *n* (%)	2 (6)	2 (5)	0

**Table 2 biomedicines-09-01444-t002:** **Differences in the molecular profile of effluent fluids collected from DCD and DBD liver grafts.** Variables were assessed by means of routine laboratory testing, gas analysis, and immunoassays. Median (25–75) is shown for each variable. Fold change was calculated by log2-trasforming the ratio of the mean of DCD over the mean of DBD for each analyte; a color-based scale was used to facilitate data visualization. Statistical significance between groups was investigated with non-parametric Wilcoxon test.

Pattern	Variable	DCD	DBD				Fold Change			*p* Value		
**Inflammation**												
	CCL2/MCP-1, pg/g	231.25 (186.4–451.9)	120.45 (57.45–238.35)				0.7			**0.048**		
	CXCL8/IL-8, pg/g	52.40 (15.40–84.70)	9.80 (4.80–14.52)				2.6			**0.002**		
	CXCL9/MIG, pg/g	104.55 (85.4–194.8)	72.02 (7.9–122.2)				0.9			**0.042**		
	IL-1b, pg/g	3.70 (1.95–17.12)	5.75 (3.15–9.45)				1.9			0.956		
	IL-6, pg/g	34.15 (18.1–56.2)	34.75 (15.1–60.56)				0.5			0.973		
	PTX3, ng/g	0.04 (0.03–0.11)	0.11 (0.04–0.21)				−0.1			0.451		
	TNF-α, pg/g	5.45 (5.30–5.60)	1.45 (0.55–2.65)				1.7			**0.037**		
	IL-10, pg/g	0.60 (0.40–1.50)	13.15 (1.00–19.00)				−3.4			**0.003**		
**Resolution, Repair, Regeneration**												
	ANGPTL3, pg/g	989.40 (523.70–1.27)	562.35 (219.10–1419.37)				0.3			0.329		
	ANGPTL4, ng/g	128.40 (106.70–212.97)	1.00 (0.50–37.70)				3.0			**0.001**		
	Galectin-9, ng/g	4.67 (3.77–6.20)	2.50 (0.97–6.87)				−0.4			0.412		
	HGF, pg/g	475.60 (367.40–919.82)	445.80 (318.75–833.00)				0.5			0.770		
**Coagulation**												
	Protein C/Factor IVX, pg/g	347.30 (193.40–514.70)	111.25 (79.17–200.35)				1.2			**0.008**		
	D-dimer, mg/g	0.99 (0.50–1.27)	1.87 (1.02–3.01)				−1.5			**0.012**		
	PAI-1, ng/g	0.11 (0.03–0.18)	0.17 (0.17–0.17)				−0.6			0.486		
**Hepatocyte injury**												
	CK18, U/g	0.63 (0.21–1.10)	0.12 (0.09–0.21)				1.0			**0.018**		
	AST, IU/g	0.45 (0.28–0.7)	0.29 (0.21–0.43)				0.6			0.221		
	ALT, IU/g	0.49 (0.22–0.76)	0.24 (0.16–0.41)				0.7			0.190		
	LDH, IU/g	0.63 (0.34–1.32)	0.64 (0.35–0.89)				0.2			0.804		
**Liver/hepatocyte metabolism**												
	BUN, mmol/g	0.005 (0.003–0.006)	0.002 (0.001–0.003)				1.3			**0.039**		
	Lac, mmol/g	0.002 (0.002–0.003)	0.002 (0.002–0.003)				0.0			0.622		
	FMN, ng/g	25.00 (16.20–36.70)	38.18 (29.60–50.70)				−0.4			0.127		
**Glycocalyx**												
	Glypican, pg/g	24.35 (13.61–61.55)	40.84 (32.76–59.04)				0.3			0.376		
	HA, ng/g	10.15 (6.10–121.95)	17.75 (8.70–28.80)				1.8			0.195		
	Syndecan-1, pg/g	637.35 (470.60–804.10)	449.30 (280.20–523.50)				0.6			0.170		
**Other**												
	K+, mmol/g	0.011 (0.010–0.015)	0.015 (0.012–0.016)				−0.3			**0.027**		
	Free hemoglobin, mcg/g	11.00 (1.70–24.55)	13.00 (5.70–21.10)				0.1			0.951		
	WBC, cells × 10^6^/g	0.70 (0.45–0.92)	0.45 (0.30–0.80)				0.8			0.305		
	RBC, cells × 10^3^/g	332.95 (181.80–580.10)	276.35 (127.20–537.30)				0.2			0.502		
**FC scale**			**−2**	**−1.5**	**−1**	**−0.5**	**0**	**+0.5**	**+1**		**+1.5**	**+2**

## Data Availability

The data presented in this study are available on request from the corresponding author.

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
