# Peer review of "Effluent Molecular Analysis Guides Liver Graft Allocation to Clinical Hypothermic Oxygenated Machine Perfusion"

_biomedicines, 2021, doi:10.3390/biomedicines9101444_

Round 1

Reviewer 1 Report

This manuscript is about an effluent molecular analysis of liver grafts. It is interesting although there are several issue to be clarified and corrected:

- EAD-yes vs EAD-no. Few results are displayed in the article. The authors must explain why in the discussion. Furthermore, the sample is very low, they must include this fact as a limitation of the study.

- Introduction: "In addition, the presence of a peculiar biomolecular signature in the effluent of DCD grafts was investigated." Since a “total” molecular screening on biomarkers has been done avoid the terms "biomolecular signature". Explain which kind of molecules have been investigated and why. 

- Introduction:  "More specifically, hepatocyte enzymes" avoid the term hepatocyte enzyme since these enzymes are also found in other human cells.

- 2.3 liver machine perfusion. It is not clear how many livers were treated with liver Machine Perfusion. The authors must indicate the number or specify that every liver covered the required characteristics. In table 1. It appears that this data could be 18, 8 and 10. The authors should report this data also in 3.2 paragraph. It is also not so clear how many subject underwent on normothermic machine perfusion (NRP). It must be specified.

-Discussion: They claim "Therefore, the decision to perfuse a specific liveror not is currently based on a few objective criteria, on the centre-specific policy and the surgeons’ individual “gut feeling”. Hence what are this criteria? what is suggesting this article? (change also “liveror” with “liver or” – correct all the refuses along the article.

-Discussion: FMN part. The authors must report what are the levels of FMN in every group. Report also the markers used to report the organ viability (ref 63,64)

Author Response

Reviewer #1

  1. EAD-yes vs EAD-no. Few results are displayed in the article. The authors must explain why in the discussion. Furthermore, the sample is very low, they must include this fact as a limitation of the study.

As highlighted by the Reviewer, the small number of liver transplantations performed in our center, as well as the low percentage of EAD observed in our case-series, affected the identification of any potential relationship between effluent biomarkers and short-term recipient outcome. This point has been further underlined in the last paragraph of the Discussion section (see page 16 of the Revised Manuscript).

This being clarified, we found higher effluent concentrations of AST and ALT transaminases together with the cytolysis marker LDH in the effluent samples collected from livers of the “EAD-yes” group, compared to grafts in the “EAD-no” group. These findings confirm and expand the results of previous investigations focused on the assessment of markers of hepatocellular damage in effluent fluids (1–8). Concerning the other variables evaluated in the present study, no statistically significant difference was observed between the “EAD-yes” and the “EAD-no” groups (see paragraph “3.4 Association with transplantation-related complications and recipient outcome”, page 12 of the Revised Manuscript). We have added a table in the Supplementary material to show these results (please see Supplementary Table 2 on pages 6-7 in the Supplementary material).

  1. Introduction: "In addition, the presence of a peculiar biomolecular signature in the effluent of DCD grafts was investigated." Since a “total” molecular screening on biomarkers has been done avoid the terms "biomolecular signature". Explain which kind of molecules have been investigated and why. 

We thank you for this comment.  To clarify the selection of biomarkers assessed in the effluent samples, we have added the sentence “In particular, we elected to assess the effluent concentration of biomarkers of hepatocellular damage, inflammation-related mediators, and DAMPs which were previously associated with brain death-induced inflammation, liver IRI, and poor transplantation outcome” in the Introduction section (please see page 2 of the Revised Manuscript).

  1. Introduction:  "More specifically, hepatocyte enzymes" avoid the term hepatocyte enzyme since these enzymes are also found in other human cells.

The manuscript has been changed accordingly.

  1. 3 liver machine perfusion. It is not clear how many livers were treated with liver Machine Perfusion. The authors must indicate the number or specify that every liver covered the required characteristics. In table 1. It appears that this data could be 18, 8 and 10. The authors should report this data also in 3.2 paragraph. It is also not so clear how many subject underwent on normothermic machine perfusion (NRP). It must be specified.

The number of livers subjected to the Machine Perfusion procedure is indicated in paragraph “3.1 Donor characteristics and liver graft allocation to the MP procedure” of the Revised version of the manuscript (page 5).

Normothermic regional perfusion (NRP) was applied to all DCD donors after death declaration and before organ procurement (paragraph “2.2. Donors, liver procurement, and preservation”, page 3), as this procedure is mandatory in DCD donation in Italy.

  1. Discussion: They claim "Therefore, the decision to perfuse a specific liveror not is currently based on a few objective criteria, on the centre-specific policy and the surgeons’ individual “gut feeling”. Hence what are this criteria?

what is suggesting this article?

change also “liveror” with “liver or” – correct all the    refuses along the article.

The criteria globally adopted for decision making on Machine Perfusion are listed in the Introduction section (page 2). On the other hand, the specific criteria used in our case-series are: expected prolonged ischemia time (>10 h), macrovescicular steatosis >30%, serum levels of hepatonecrosis markers exceeding 4 times the reference range (please see paragraph “2.3. Liver machine perfusion (MP)”, page 3).

Our proof-of-concept study provides significant results suggesting that effluent molecular analysis can assist surgeons in the clinical decision making on MP procedure. In fact, effluent biomarker evaluation can provide additional objective criteria to reduce the degree of uncertainty and subjectivity in the selection process of livers requiring or not further assessment with D-HOPE before transplantation. 

The spelling errors were corrected accordingly. Thank you for this suggestion.

  1. Discussion: FMN part. The authors must report what are the levels of FMN in every group. Report also the markers used to report the organ viability (ref 63,64)

We thank the reviewer for this important point.

The levels of FMN in the effluent found in the different groups were:  MP-no vs MP-yes: 45.80 [37.75-51.75] ng/g vs 24.60 [15.70-33.58] ng/g, p=0.001 (Figure 2);  DCD vs DBD: 25.00 [16.20-36.70] ng/g vs 38.18 [29.60-50.70] ng/g, p=0.127 (Table 2);  EAD-no vs EAD-yes: 33.50 [20.60-50.32] ng/g vs 37.60 [33.10-52.00] ng/g, p=0.246 (Supplementary table 2).

Data related to other markers of cell damage, including AST, ALT, and LDH were added in the paragraph “3.3.2. Effluents of livers referred to MP procedure show a peculiar molecular signature” of the Revised version of the manuscript (page 7). Thank you for pointing out that this information was missing.

Reviewer 2 Report

The study by Lonati et al is a sound contribution to the upcoming HOPE era. The paper is well written and comprehensive. The quality of the illustrations could be improved in some places (F.e. Figure 1).

Additionally the authors investigated a plethora of biomarkers. However no correlation between biomarkers effluent analysis and clinical outcome.  Why do the authors believe that this method can help to improve the clinical outcome of LT or represents a further step towards individualized medicine? Would the authors have retrospectively rejected some of the transplanted organs if they had the analysis results earlier? If yes, then the donor pool would rather shrink, than expand. Albeit this ambiguity, which of the markers the authors think are the most important in clinics and should be added in a real-time evaluation panel? Do the authors have already started to select/reject livers by their biomarker fingerprint? 

Author Response

Reviewer #2

  1. The quality of the illustrations could be improved in some places (F.e. Figure 1).

Figure 1 has been improved. Thank you very much for pointing this out.

  1. The authors investigated a plethora of biomarkers. However no correlation between biomarkers effluent analysis and clinical outcome.  Why do the authors believe that this method can help to improve the clinical outcome of LT or represents a further step towards individualized medicine?

We thank the reviewer for this important point.

Although the general purpose of this research includes improvement of clinical outcome of LT, the specific aim in this work was on providing a more informed capacity in the decision making on the D-HOPE referral.

The results of our research support the application of effluent analysis as a complementary method to assist surgeons in the decision making on the MP procedure. In fact, effluent biomarkers provide significant information to assess the degree of the factual injury experienced by each liver graft (1–18). The idea is that this procedure could help assessment  of the detrimental changes occurring during the transplantation process -i.e., donor management, surgery/procurement, cold preservation- in the hepatic tissue. We hope that our proof-of-concept study will encourage not only the adoption of tailored selection of the grafts really requiring further evaluation with D-HOPE, but also the reduction of futile machine perfusions. This would lead to improved donor-recipient matching, better post-transplant results, and optimized healthcare resource utilization.

Further studies are required to provide conclusive results with a wider range of donor risk profiles, not only to determine predictor thresholds, but also to support our results with additional evidence from a higher number or organs and transplantations (ideally from various centres).

  1. Would the authors have retrospectively rejected some of the transplanted organs if they had the analysis results earlier? If yes, then the donor pool would rather shrink, than expand.

We thank the reviewer for this comment and question.

Effluent profiling can be applied as an ancillary method offering significant information to improve liver graft allocation to D-HOPE. However, effluent analysis cannot be the basis to either accept or reject liver grafts because it cannot provide information on the biological events elicited in the liver tissue during D-HOPE. Consequently, the results of the present study would have not be used to reject some of the transplanted organs and, therefore it would have not shrinked the donor pool.

  1. Albeit this ambiguity, which of the markers the authors think are the most important in clinics and should be added in a real-time evaluation panel? Do the authors have already started to select/reject livers by their biomarker fingerprint? 

Clinical translation of effluent and D-HOPE perfusate biomarker analysis is currently ongoing at our Transplant Centre. Of note, on-line evaluation of FMN can be readily implemented, due to the auto-fluorescent characteristics of this molecule, (19), while real-time measurement of mediator concentration (cytokine, chemokines, and DAMPs) is now feasible using innovative biotechnology methods recently introduced in the clinical practice (20–22). Our research team is investigating the molecular profile of both effluent fluids and D-HOPE perfusates in order to identify which biomarkers can best describe liver quality and viability.
